# LLM-powered Data Augmentation for Enhanced Crosslingual Performance

**Chenxi Whitehouse**[1,3,*]   **Monojit Choudhury**[2]   **Alham Fikri Aji**[3]
[1]City, University of London   [2]Microsoft   [3]MBZUAI
chenxi.whitehouse@city.ac.uk
monojitc@microsoft.com   alham.fikri@mbzuai.ac.ae

## Abstract

This paper explores the potential of leveraging Large Language Models (LLMs) for data augmentation in multilingual commonsense reasoning datasets where the available training data is extremely limited. To achieve this, we utilise several LLMs, namely Dolly-v2, StableVicuna, ChatGPT, and GPT-4, to augment three datasets: XCOPA, XWinograd, and XStoryCloze. Subsequently, we evaluate the effectiveness of fine-tuning smaller multilingual models, mBERT and XLMR, using the synthesised data. We compare the performance of training with data generated in English and target languages, as well as translated English-generated data, revealing the overall advantages of incorporating data generated by LLMs, e.g. a notable 13.4 accuracy score improvement for the best case. Furthermore, we conduct a human evaluation by asking native speakers to assess the naturalness and logical coherence of the generated examples across different languages. The results of the evaluation indicate that LLMs such as ChatGPT and GPT-4 excel at producing natural and coherent text in most languages, however, they struggle to generate meaningful text in certain languages like Tamil. We also observe that ChatGPT falls short in generating plausible alternatives compared to the original dataset, whereas examples from GPT-4 exhibit competitive logical consistency. We release the generated data at https://github.com/mbzuai-nlp/Gen-X.

## 1 Introduction

The success of NLP models greatly depends on the availability and quality of training data. This poses a significant challenge for multilingual NLP, as data for languages other than English is typically limited (Ponti et al., 2019; Joshi et al., 2020; Whitehouse et al., 2022). An approach to address the data scarcity challenge is through zero-shot crosslingual transfer or multitask training, in which a

model is trained across data of diverse tasks and languages, exhibiting the capability to handle unseen tasks, particularly in larger models (Artetxe and Schwenk, 2019; Nooralahzadeh et al., 2020; Huang et al., 2021). However, when aiming for task-specific objectives, a smaller, fine-tuned model dedicated to that particular task often outperforms larger general-purpose, zero-shot models. In addition, a smaller task-specific model is more practical and cost-effective for training and deployment. Nevertheless, developing a powerful task-specific model becomes challenging in the absence of training data (Lauscher et al., 2020).

Conversely, recent powerful Large Language Models (LLMs) excel at handling general instructions and have shown promise in data generation tasks (Wang et al., 2023). In this work, we leverage LLMs to generate synthetic data for various multilingual commonsense reasoning tasks, XCOPA (Ponti et al., 2020), XWinograd (Tikhonov and Ryabinin, 2021), and XStoryCloze (Lin et al., 2022), where the training data is limited even for English (see Table 1). To augment the training data, we provide LLMs with instructions and examples from the original training data, prompting them to generate new and diverse examples. We explore the generation of synthetic data in English using different LLMs, including open-source models like Dolly-v2[1] and StableVicuna[2], as well as ChatGPT and GPT-4. Although the weights and capabilities of the latter two models remain undisclosed, we explore them as they extend the capability of generating texts in languages beyond English.

We develop task-specific models by fine-tuning multilingual pre-trained language models, namely mBERT (Devlin et al., 2019) and XLM-R (Conneau et al., 2020), using the generated data. We then compare their performance against models trained on a limited set of human-created data in the

---

*Work conducted while visiting MBZUAI.

[1]https://github.com/databrickslabs/dolly
[2]https://github.com/Stability-AI/StableLM

| DATASET | Train | | Validation | | Test | |
|---|---|---|---|---|---|---|
| | EN | XX | EN | XX | EN | XX |
| XCOPA | 400 | 0 | 100 | 100 | 500 | 500 |
| XWinograd | 1858 | 0 | 233 | 0 | 233 | 424 |
| XStoryCloze | 300 | 300 | 60 | 60 | 1511 | 1511 |

Table 1: Number of examples available in XCOPA, XWinograd, and XStoryCloze. XX denotes the average number of non-English examples per language. Since a validation split is not specified in XStoryCloze, we take 60 random examples from the train split for validation. XWinograd has no train/validation/test split, and we follow an 80/10/10 split for the experiments.

target language whenever available, and otherwise through zero-shot transfer learning from manually created English training data. Our experiments demonstrate that training the models with *relatively large* synthetically generated datasets yields better performance than training with *limited* manually-created datasets. This finding empirically confirms the utility of synthetic data generated by LLMs for improving downstream task-specific models.

We expand the multilingual data synthesis using ChatGPT and GPT-4 on XCOPA and find that generating multilingual datasets generally surpasses the effectiveness of the zero-shot cross-lingual transfer. We further assess the quality of the generated dataset in different languages by asking native speakers to evaluate the naturalness and logical soundness of the generated dataset compared to the human-written examples. The annotation results reveal that while ChatGPT and GPT-4 successfully generate natural text in most languages, they struggle with generating understandable text in certain languages such as Tamil. Moreover, a noticeable gap is observed in terms of commonsense coherence when comparing ChatGPT-generated data to human-constructed data. On the other hand, GPT-4 significantly narrows this difference.

To summarise, our work has the following key contributions:

- Augmenting three low-resource, multilingual commonsense reasoning datasets by leveraging and prompting four LLMs;
- Fine-tuning smaller models, mBERT and XLMR, using the synthesised data and showcasing the practical value of the LLM-generated data;
- Performing an extensive analysis of the effects of various target languages in data generation and scaling, as well as a human evaluation of

the naturalness and logical coherence of the data generated in various languages;
- Releasing the synthesised datasets for public use and reproducibility.

## 2 Related Work

**Multilingual and Low-Resource NLP**

Recently, there has been increased attention on expanding NLP beyond English, including the development of multilingual models (Devlin et al., 2019; Conneau et al., 2020; Xue et al., 2021; Scao et al., 2022) as well as the creation of benchmarks to address multilingual challenges (Conneau et al., 2018; Artetxe et al., 2020; Adelani et al., 2021; Winata et al., 2023). Among the prevailing challenges faced across various languages, a common theme is the scarcity of available data.

Consequently, when data is lacking, one approach is to employ zero-shot cross-lingual transfer. Studies conducted by Winata et al. (2023) have demonstrated the effectiveness of zero-shot cross-lingual transfer for related languages. Additionally, Muennighoff et al. (2023) show that models fine-tuned only with English instruction data are capable of understanding multilingual instructions. In this work, we are tackling a similar scenario where the availability of data is limited.

**Multilingual Data Augmentation**

Lauscher et al. (2020) show that few-shot can drastically increase the cross-lingual performance of small models, proving that multilingual data augmentation is an effective strategy. A series of works try to predict the cross-lingual accuracy of models through measurements and modelling (Xia et al., 2020), and study strategies for multilingual data augmentation, such as choosing the transfer languages (Lin et al., 2019), and predicting multilingual few-shot accuracy leading for optimal data augmentation approaches (Srinivasan et al., 2022).

Many works focus on synthetic data augmentation for code-mixing, including utilising linguistic theories (Lee et al., 2019; Pratapa et al., 2018), machine translation models (Tarunesh et al., 2021), parallel corpus and Wikipedia (Winata et al., 2019; Whitehouse et al., 2022), and employing ChatGPT (Dai et al., 2023). Our work explores data augmentation on multilingual commonsense datasets with powerful instruction-tuned LLMs.

| XCOPA | XWINOGRAD | XSTORYCLOZE |
|---|---|---|
| 👤 We are collecting more examples for the COPA dataset which will be used to test a system's ability of Commonsense Causal Judgments. The format of the data: A premise: a statement of something that happened, and two choices that could plausibly {*occur as the result* / *be the cause*} of the premise. The correct choice is the alternative that is more plausible than the wrong choice. Here are $n$ examples in {language}: Example 1: Premise: The man wanted to save money. What happened as a result? Correct choice: He cut back on making frivolous purchases. Wrong choice: He withdrew money from his savings account. … Example $n$: … Based on the examples above, generate $m$ new examples in {language}. | 👤 We are collecting more examples for the Winograd Schema Challenge. Each example has a short sentence that contains two noun phrases and one pronoun replaced by "_", and the challenge is to determine the referent of the pronoun, which can only be inferred from the context. Here are $n$ examples of the data: Example 1: Sentence: Harley hides from Dyna because _ is scary. Who/What is scary? Correct answer: Dyna. Wrong answer: Harley. … Example $n$: … Based on the examples above, generate $m$ new examples. Both noun phrases in each example can be males, females, inanimate objects, or groups of people or objects. There should only be one "_" in the sentence. The correct and wrong answer should be one of the noun phrases mentioned in the sentence. | 👤 We are collecting more examples for a story cloze dataset. Each example consists of a 4-sentence story, one correct ending sentence which is a plausible continuation of the story, and one wrong ending sentence which is logically inconsistent with the context. Here are $n$ examples of the data: Example 1: Sent-1: Tina is very tired every single morning. Sent-2: She does not get enough sleep because of her two jobs. Sent-3: Tina decides to quit one of the jobs. Sent-4: She now gets enough sleep to function everyday. Correct ending: Tina is well rested. Wrong ending: Tina is more tired than ever before. … Example $n$: … Based on the examples above, provide $m$ new similar examples. Requirements: 1) the story should read like a coherent story, with a specific beginning and ending, where something happens in between 2) both ending sentences should be entirely reasonable, realistic and sensible when read in isolation, and 3) both ending sentences should follow up the story by sharing at least one of the characters of the story. |
| 💬 Premise: The politician made a controversial statement. What happened as a result? Correct choice: The politician faced criticism from the media. Wrong choice: The politician's approval ratings increased. 💬 Premise: 我裤子口袋里的钥匙不见了。 What was the cause? Correct choice: 这个口袋上有一个洞。 Wrong choice: 裤子是新的。 | 💬 Sentence: Sam gave Andrew the book because _ had already read it. Who/What had already read the book? Correct answer: Sam. Wrong answer: Andrew. 💬 Sentence: The dog chased the cat , but _ was too fast. Who/What was too fast? Correct answer: the cat. Wrong answer: The dog. | 💬 Sent-1: Jordan was a high school student who wanted to become a doctor. Sent-2: He spent all his free time studying biology and chemistry. Sent-3: One day, his school hosted a science fair competition. Sent-4: Jordan's project won first place. Correct ending: Jordan went on to study medicine in college. Wrong ending: Jordan gave up his dream of becoming a doctor. |

Table 2: Examples of instructions and LLM-responses (ChatGPT) for XCOPA, XWinograd, and XStoryCloze.

# 3 Dataset Augmentation

Our experiments use XCOPA, XWinograd, and XStoryCloze, which are selected due to (1) the limited availability of training data and (2) commonsense reasoning datasets present greater challenges for data synthesis. Table 1 summarises the statistics of the three datasets.

**XCOPA** is a cross-lingual Choice of Plausible Alternatives dataset that translates and re-annotates the validation and test sets of English (EN) COPA (Roemmele et al., 2011) into 11 target languages (ET: Estonian, HT: Haitian Creole, ID: Indonesian, IT: Italian, QU: Quechua, SW: Swahili, TA: Tamil, TH: Thai, TR: Turkish, VI: Vietnamese, and ZH: Chinese).[3] Each instance consists of a premise, a question (*cuase*/*result*), and two alternatives. The task is to predict the more plausible alternative.

**XWinograd** expands the original English Winograd Schema Challenge (WSC) (Levesque et al., 2012) to five other languages (FR: French, JA: Japanese, PT: Portuguese, RU: Russian, and ZH),[4] which consists of pronoun resolution problems aiming to evaluate the commonsense reasoning ability of a machine. Given a statement with two noun phrases and a pronoun, the challenge of WSC is to determine the referent of the pronoun, which can only be inferred from the context.

**XStoryCloze** is collected by Lin et al. (2022), where the validation split of the original English StoryCloze dataset (Mostafazadeh et al., 2016) is translated into 10 other typologically diverse languages (RU, ZH, ES: Spanish, AR: Arabic, HI: Hindi, ID, TE: Telugu, SW, EU: Basque, and MY: Burmese). Each example consists of a four-sentence commonsense story, a correct ending, as well as a wrong ending.

## 3.1 LLMs for Data Generation

Our preliminary experiments reveal that language models that are specifically fine-tuned on downstream NLP tasks, such as BLOOMZ (Scao et al., 2022) and Flan-T5 (Chung et al., 2022), struggle to follow the complex instructions. Conversely, more recent LLMs such as Dolly-v2, StableVicuna, ChatGPT, and GPT-4, which are designed to handle more intricate and general-purpose instructions, have demonstrated success in following our instructions for data generation. ChatGPT and GPT-4 also

---

[3] https://huggingface.co/datasets/xcopa
[4] https://huggingface.co/datasets/Muennighoff/xwinograd

stand out with the capability of generating examples in non-English languages.

We explore synthetic data generation with the four aforementioned LLMs, balancing between open-access models and closed models (see §5.1). Specifically, we use dolly-v2-12b,[5] which is derived from EleutherAI's Pythia-12b (Biderman et al., 2023) and fine-tuned on a ~15K instructions generated by Databricks employees; and StableVicuna-13B, an RLHF (reinforcement learning from human feedback) fine-tuned Vicuna model on various conversational and instructional datasets - Vicuna is an open-source LLaMA model (Touvron et al., 2023a) fine-tuned on user-shared conversations collected from ShareGPT.[6]

## 3.2 Instructions and Responses

We utilise LLMs to generate synthetic examples for all datasets by prompting them. We construct instructions using the descriptions from the dataset papers as a reference and provide LLMs with some examples, randomly sampled from the *train (+validation)* split of the original dataset, then ask LLMs to generate similar data points. We experiment with various instructions and evaluate the synthesised data on a smaller scale, update the instructions based on the errors, and then choose the best instruction to generate the final datasets.

The final instructions and responses are in Table 2. Our data generation process comprises the following key steps: (1) We establish the desired total number of examples to generate. This quantity can be determined by various factors such as budget constraints, a fixed ratio concerning the original dataset, etc. (2) We proceed to generate examples through the following iterative process: (a) To ensure diversity,[7] we randomly sample a set of $n$ examples from the training datasets. (b) We append these sampled examples to the instructions and prompt the model to generate an additional set of $m$ new examples. (c) Afterwards, we perform post-processing and only add valid and unique examples to the generated set. Typically, the values of $n$ and $m$ are set to 5 to 10.

We focus on a fixed-budget scenario and first generate a total of 3-4K data points for each dataset with LLMs. LLMs tend to generate fewer samples than requested or inconsistent output in invalid for-

| Model | XCOPA | XWinograd | XStoryCloze |
|---|---|---|---|
| DOLLY-V2 | 41.6% | 22.4% | 41.2% |
| STABLEVICUNA | 36.1% | 33.8% | 36.1% |
| CHATGPT | 86.4% | 43.8% | 77.6% |
| GPT-4 | 89.7% | 85.0% | 89.3% |

Table 3: Generation Success Rate in English (valid examples obtained / total examples requested) with different LLMs on the three datasets.

mats. We report the success rate for different LLMs on the three datasets in Table 3, which indicates that GPT-4 has the most robustness.

Among the datasets, LLMs have the lowest generation success rate for XWinograd, which is more challenging. XWinograd requires both answers to be from the generated sentence, with only one pronoun being replaced. In addition, we observed pronoun inconsistency in the generated XWinograd data. Despite the requirement for interchangeable pronouns in the options, models frequently fail to comply. For example, "The dog bit the mailman because _ entered the yard." is generated by ChatGPT with the options 'The dog'" or "the mailman", however, "_" in the sentence cannot be replaced by the same pronoun for the given two options, hence it may make the task easier and the example is considered suboptimal. We keep those instances in the dataset and discuss further in §6.1.

## 4 Experimental Setups

We first generate synthetic English examples for XCOPA, XWinograd, and XStoryCloze, with Dolly-v2, StableVicuna, ChatGPT, and GPT-4. The size of the final filtered synthesised data for the three datasets is 3.7k, 2K, and 1.7K, respectively. We then fine-tune mBERT, XLMR-base, and XLMR-large with the synthesised data and compare the zero-shot cross-lingual transfer performance across different languages, where we use the original validation set in target languages.

For XCOPA, we additionally experiment with generating data points directly in non-English languages, by providing examples in the target language and specifying the language desired for the generated data (see Table 2). However, since no examples for *cause* are included in TH and TR train/validation data (they do appear in the test split), we do not generate XCOPA for the two languages. We use ChatGPT and GPT-4 for multilingual synthetic data generation, as both Dolly-v2

---

[5]Model details are included in Appendix A.

[6]https://github.com/lm-sys/FastChat

[7]An analysis of the diversity of the generation as well as topic coverage is included in Appendix B.

| Fine-tuned Model | LLM for Generation | XCOPA | | | XWINOGRAD | | | XSTORYCLOZE | | |
|---|---|---|---|---|---|---|---|---|---|---|
| | | $ORI_{400}$ | $GEN_{3.7k}$ | $O+G_{4.1k}$ | $ORI_{1.8k}$ | $GEN_{2k}$ | $O+G_{3.8k}$ | $ORI_{300}$ | $GEN_{1.7k}$ | $O+G_{2k}$ |
| mBERT | DOLLY-V2 | 47.9 | 53.3 ↑5.4 | 54.0 ↑6.1 | 52.9 | **59.6** ↑6.7 | **59.3** ↑6.4 | 65.0 | **68.7** ↑3.7 | 68.1 ↑3.1 |
| | STABLEVICUNA | 47.9 | 52.9 ↑5.0 | 54.7 ↑6.8 | 52.9 | 53.7 ↑0.8 | 58.5 ↑5.6 | 65.0 | 64.6 ↓0.4 | 67.3 ↑2.3 |
| | CHATGPT | 47.9 | 55.0 ↑7.1 | 54.1 ↑6.2 | 52.9 | 56.0 ↑3.1 | 58.3 ↑5.4 | 65.0 | 64.3 ↓0.7 | 68.3 ↑3.3 |
| | GPT-4 | 47.9 | **56.4** ↑8.5 | **57.2** ↑9.3 | 52.9 | 54.9 ↑2.0 | 57.5 ↑4.6 | 65.0 | 68.0 ↑3.0 | **69.8** ↑4.8 |
| XLMR-Base | DOLLY-V2 | 54.8 | 58.1 ↑3.3 | 58.1 ↑3.3 | 53.5 | 56.5 ↑3.0 | 66.3 ↑12.8 | 73.0 | 75.8 ↑2.8 | 76.5 ↑3.5 |
| | STABLEVICUNA | 54.8 | 57.6 ↑2.8 | 59.3 ↑4.5 | 53.5 | 59.0 ↑5.5 | 66.0 ↑12.5 | 73.0 | 69.6 ↓3.4 | 74.2 ↑1.2 |
| | CHATGPT | 54.8 | 58.2 ↑3.4 | 59.4 ↑4.6 | 53.5 | 62.7 ↑9.2 | 65.9 ↑12.4 | 73.0 | 67.4 ↓5.6 | 74.5 ↑1.5 |
| | GPT-4 | 54.8 | **62.7** ↑7.9 | **63.0** ↑8.2 | 53.5 | **63.3** ↑9.8 | **66.9** ↑13.4 | 73.0 | **74.6** ↑1.6 | **79.3** ↑6.3 |
| XLMR-Large | DOLLY-V2 | 63.0 | 58.6 ↓4.4 | 65.0 ↑2.0 | 80.1 | **76.9** ↓3.2 | 83.1 ↑3.0 | 85.0 | 84.8 ↓0.2 | 86.4 ↑1.4 |
| | STABLEVICUNA | 63.0 | 64.4 ↑1.4 | 68.7 ↑5.7 | 80.1 | 68.2 ↓11.9 | 82.0 ↑1.9 | 85.0 | 74.6 ↓10.4 | 84.8 ↓0.2 |
| | CHATGPT | 63.0 | 64.6 ↑1.6 | 68.1 ↑5.1 | 80.1 | 73.2 ↓6.9 | 83.2 ↑3.1 | 85.0 | 77.3 ↓7.7 | 85.8 ↑0.8 |
| | GPT-4 | 63.0 | **72.1** ↑9.1 | **72.2** ↑9.2 | 80.1 | 76.4 ↓3.7 | **83.5** ↑3.4 | 85.0 | **86.0** ↑1.0 | **88.4** ↑3.4 |

Table 4: Comparison of Average Accuracy across all languages for mBERT, XLMR-Base, and XLMR-Large on XCOPA, XStoryCloze, and XWinograd. Training datasets include *ORI* (original EN data), *GEN* (LLM-generated EN data), and *O+G* (both), with the number of examples used for training indicated by the subscripts. The best results obtained with the same amount of training data are highlighted in bold. Green and red subscripts denote improvement and decline in performance compared to the baseline (*ORI*). See per language results in Appendix D.

and StableVicuna exhibit limitations in effectively generating multilingual text. The size of the multilingual synthesised data is ∼3.6K in each language.

We fine-tune models on all datasets as multiple-choice tasks[8] by searching best learning rate from $\{5e^{-6}, 10e^{-6}\}$, and batch size from $\{8, 16, 32\}$. All the fine-tuning experiments are conducted on a single 40G A100. For generating data with Dolly-v2 and StableVicuna, we use 2×40G A100.

## 5 Results and Discussion

This section presents the main results of fine-tuned models on the three datasets and compares performance with generated data in different LLMs, languages, and scales.

### 5.1 General Result

Table 4 presents the average accuracy of fine-tuned mBERT, XLMR-Base, and XLMR-Large models across all languages on the three datasets. The models are trained using original data (*ORI*), different LLM-generated data (*GEN*), as well as a combination of both sources (*O+G*) in English.

Across different datasets, LLMs, and fine-tuned models, consistent improvements are observed when using both original and LLM-generated data. Among the models, Dolly-v2 performs the best on Xingorad when fine-tuned on mBERT, while

---

[8]In our preliminary experiments, we find that formulating XWinograd as a binary text classification results poorly, in line with the observation from Liu et al. (2020) that the task formulation is essential to the performance of Winograd.

GPT-4 achieves the highest accuracy in other settings. The most significant improvement is shown in XWinograd with XLMR-Base, where the addition of an extra 2k datapoints leads to an average accuracy enhancement of 12.8 compared to the baseline, across all four LLMs.

When using only LLM-generated data, smaller models like mBERT and XLMR-Base generally outperform the baseline. However, with XLMR-Large, which achieves stronger baselines. e.g. >80 in XWinograd and XStoryCloze, the accuracy remains similar or even worse compared to using the original data. GPT-4-generated data demonstrates the best robustness but still experiences a decline in performance in XWinograd when the generated data size is similar to the original data. This highlights the challenges of generating data at a human-level quality.

### 5.2 Multilingual Data Generation

We investigate whether the synthetically generated multilingual dataset outperforms training solely in English. We choose the XCOPA dataset and explore two settings: synthetic multilingual data by asking LLMs to generate responses in the target languages directly and translating the English-generated data to target languages with Google Translate API. We exclude Dolly-v2 and StableVicuna due to their limited effectiveness in generating non-English text. Although GPT-4 exhibits the most promising performance, it is significantly costlier compared to ChatGPT. Therefore, we also

| Fine-tuned | LLM | Training data | AVG | EN | ET | HT | ID | IT | SW | TA | VI | ZH |
|---|---|---|---|---|---|---|---|---|---|---|---|---|
| mBERT | BASELINE | $ORI$ | 47.2 | 53.8 | 44.2 | 48.6 | 47.2 | 46.2 | 45.4 | 48.4 | 43.6 | 47.4 |
| | CHATGPT | $GEN_{EN} + ORI$ | 54.6 | 59.6 | 56.4 | 53.6 | 53.8 | 51.4 | 51.6 | 50.4 | 55.0 | 59.2 |
| | | $GEN_{XX} + ORI$ | 56.8 | 59.6 | 58.8 | 54.6 | 56.2 | 61.2 | 54.6 | 53.6 | 52.0 | 60.2 |
| | | $GEN_{EN}^{Trans} + ORI$ | 58.7 | 59.6 | 59.8 | 58.2 | 62.8 | 61.0 | 52.6 | 56.8 | 58.2 | 59.4 |
| | GPT-4 | $GEN_{EN} + ORI$ | 59.3 | 72.6 | 58.8 | 53.0 | 62.0 | 61.0 | 50.0 | 54.0 | 57.6 | 64.6 |
| | | $GEN_{XX} + ORI$ | 61.8 | 72.6 | 61.2 | 58.2 | 62.2 | 66.4 | 57.4 | 53.4 | 63.0 | 61.8 |
| | | $GEN_{EN}^{Trans} + ORI$ | 62.6 | 72.6 | 58.6 | 55.2 | 65.6 | 65.4 | 53.8 | 62.6 | 64.6 | 65.4 |
| XLMR-Base | BASELINE | $ORI$ | 55.6 | 57.6 | 54.6 | 50.6 | 59.6 | 54.8 | 55.0 | 53.4 | 54.8 | 59.6 |
| | CHATGPT | $GEN_{EN} + ORI$ | 59.8 | 63.8 | 61.6 | 51.6 | 62.6 | 59.8 | 51.6 | 60.4 | 64.8 | 62.0 |
| | | $GEN_{XX} + ORI$ | 59.9 | 63.8 | 60.6 | 55.0 | 64.6 | 59.6 | 54.6 | 56.4 | 59.6 | 64.8 |
| | | $GEN_{EN}^{Trans} + ORI$ | 61.1 | 63.8 | 60.0 | 58.0 | 65.0 | 60.8 | 53.8 | 60.2 | 62.6 | 66.0 |
| | GPT-4 | $GEN_{EN} + ORI$ | 63.6 | 69.6 | 63.8 | 51.2 | 67.2 | 62.4 | 58.4 | 63.8 | 66.8 | 69.4 |
| | | $GEN_{XX} + ORI$ | 64.0 | 69.6 | 62.2 | 56.2 | 68.6 | 63.8 | 57.8 | 61.2 | 66.8 | 70.0 |
| | | $GEN_{EN}^{Trans} + ORI$ | 63.9 | 69.6 | 61.6 | 56.6 | 68.4 | 65.2 | 58.2 | 60.2 | 66.0 | 69.6 |
| XLMR-Large | BASELINE | $ORI$ | 64.4 | 71.4 | 62.8 | 51.4 | 69.0 | 65.8 | 60.6 | 62.0 | 69.4 | 66.8 |
| | CHATGPT | $GEN_{EN} + ORI$ | 69.5 | 76.4 | 69.8 | 48.2 | 76.0 | 72.8 | 63.4 | 67.8 | 73.4 | 77.8 |
| | | $GEN_{XX} + ORI$ | 65.2 | 76.4 | 62.4 | 55.2 | 75.0 | 62.2 | 58.2 | 55.4 | 66.2 | 76.2 |
| | | $GEN_{EN}^{Trans} + ORI$ | 67.0 | 76.4 | 60.0 | 59.6 | 66.2 | 66.6 | 59.0 | 64.8 | 74.8 | 75.6 |
| | GPT-4 | $GEN_{EN} + ORI$ | 73.7 | 84.6 | 70.4 | 50.0 | 80.8 | 80.2 | 65.8 | 72.8 | 78.4 | 80.4 |
| | | $GEN_{XX} + ORI$ | 74.6 | 84.6 | 77.0 | 56.0 | 82.2 | 77.0 | 65.0 | 73.8 | 76.2 | 80.0 |
| | | $GEN_{EN}^{Trans} + ORI$ | 74.1 | 84.6 | 74.2 | 57.2 | 82.0 | 77.4 | 62.2 | 75.0 | 74.4 | 79.6 |

Table 5: Accuracy on XCOPA. $ORI$ corresponds to the original data, $GEN_{EN}$ and $GEN_{XX}$ represents data generated in English and target languages. $Trans$ denotes translations of the English-generated data. We show languages that are available in all settings. Improvement and decline in performance are represented with green and red shadows.

consider using ChatGPT as a contrasting experiment under resource-constrained conditions.

Table 5 shows the results for the languages that are available for all settings, excluding TR and TH (unavailable for LLM-generation, refer to §4), and QU (not supported by the Google Translate API). We can see the impact of the generated data varies across different fine-tuned models and languages, aligning with the findings of Kumar et al. (2022). Training on GPT-4 synthesised data displays consistent improvement across all scenarios and languages, except the zero-shot cross-lingual result on HT with XLMR-Large.

More fluctuating results can be observed with ChatGPT-generated data. A comparison between $GEN_{EN} + ORI$ and $GEN_{XX} + ORI$ indicates that utilising data generated in target languages generally leads to improved performance with GPT-4 generated data, as well as in base models with ChatGPT-generated data. However, for XLMR-Large, employing ChatGPT-generated data in target languages mostly yields negative outcomes. In languages such as TA and VI, training on generated data in the target languages results in more performance degradation compared to zero-shot cross-lingual transfer. This suggests that ChatGPT

performs worse in those languages than XLMR-Large (Ahuja et al., 2023).

Translating the English dataset generally shows overall better results than training on the data generated directly in the target languages, with the exception of XLMR-Large with GPT-4. For SW, XLMR models fined-tuned with ChatGPT-generated data exhibit performance decline in most cases, even when the English-generated data benefits all other languages. This observation suggests that XLMR struggles with SW. In §6.1 we select TA, SW, and the two best languages, ID and ZH, along with EN, for human evaluation.

Additionally, we conduct experiments adding Target Languages in Validation (TLV). This only results in minor variations in the performance, consistent with the findings of Ponti et al. (2020). We include the full results in Table 11 in Appendix D.

### 5.3 Dataset Scaling Up

We now investigate the impact of training on a larger scale of generated data on model performance. We focus on the XCOPA dataset and expand the generated data with ChatGPT (more budget-efficient) to 28.6k examples in English. We also compare the results of zero-shot cross-lingual

| Model | $GEN_{EN} + ORI_{EN}$ | | $GEN_{EN}^{Trans} + ORI_{EN}$ | |
|---|---|---|---|---|
| | 3.7K | 28.6K | 3.7K | 28.6K |
| mBERT | 54.3 | 56.0 | 58.0 | **60.1** |
| XLMR-Base | 60.1 | **61.8** | 61.2 | 61.7 |
| XLMR-Large | 69.7 | **72.4** | 67.2 | 71.4 |

Table 6: Accuracy on XCOPA when scaling up the generated data to over 28K with ChatGPT. We report average results on all XCOPA languages excl. QU, since it is not available with the Google Translate API.

transfer with translating the English-generated data to target languages.

The results in Table 6 demonstrate the positive impact of scaling up the generated data on model performance. Particularly, XLMR-Large exhibits the most significant improvement.

Furthermore, we conduct experiments on generating data with a fixed ratio of the original datasets and the results are included in Appendix C.

# 6 Human Evaluation

To better evaluate the quality of the generated datasets and compare them with the human-created data, we ask native speakers to annotate the multilingual data generated by ChatGPT and GPT-4.

For each dataset, we first select 50 generated examples in English, and then request two annotators to evaluate the examples in two categories: (1) **Text Naturalness**. The annotators are asked to choose one of the following options for each example: "the text sounds natural", "the text sounds awkward but understandable", or "the text is not understandable", and (2) **Logic Soundness**. This category focuses on the commonsense aspect of the examples. The annotators are required to select the most appropriate description from: "the correct option is (clearly) more plausible", "both options are equally plausible", "both options are implausible", or "the wrong option is actually more plausible". We only ask the annotators to evaluate the logic if the text is at least understandable.

For XWinograd, we introduce an additional evaluation criterion. Annotators are asked to determine whether the two noun phrases in the examples can be replaced by the same pronoun (refer to §3.2). For XCOPA, we extend the annotations to non-English languages, where we choose the two languages that demonstrate the most notable improvement, namely ZH and ID, as well as the two languages that exhibit the least improvement or regres-

sion in performance with ChatGPT-generated data, namely TA and SW (see Table 5). In addition to the original examples and the generated examples in the target languages, we include 50 examples that are translated from the same English-generated examples (that were selected for annotation).

To ensure impartiality, all the examples are shuffled, and the annotators are not provided with information regarding the source of the examples (human-created, LLM-generated, or translated).

## 6.1 Text Naturalness

Figure 1 presents the annotation results for XCOPA, averaged from two annotators for each language. For Text Naturalness, we can see that in EN, ID, ZH, and SW, both ChatGPT and GPT-4 achieved higher naturalness than the original dataset. This is particularly prominent in ID, revealing the fluency issue in the original ID data in XCOPA, which is also confirmed by a native speaker.

**Issue with Tamil**

In contrast, the performance of the TA dataset is surprisingly low, with a majority of examples classified as "not understandable." Upon consulting language experts, we have identified several main issues in Tamil, including (1) the insertion of redundant words with the same meaning, such as "I will retry to try it again" (2) verb agreement errors, and (3) the presence of uncommon and out-of-context words.

It is worth noting that generating Tamil using GPT-4 is both slow and costly. We suspect that the tokenizer for Tamil, as well as similar languages like Telugu and Kannada, are poorly trained, resulting in unusable generation in those languages. While the low quality of the generated data could explain the significant decline in the performance of the XLMR-Large model when trained on ChatGPT-generated data in Tamil, intriguingly, models trained on Tamil data generated by GPT-4 show improvement over the baselines.

To further investigate this issue, we conduct an experiment where we fine-tune the models using only five examples from the TA examples generated by GPT-4 that are identified as natural and sound by the annotators. The improvement on mBERT under this setting is 50% of the total improvement seen with the entire 3.6K TA examples. For XLMR-base and XLMR-large, 15% and 3% of the total improvement can be observed, respectively. Considering that the estimated number of correct samples in

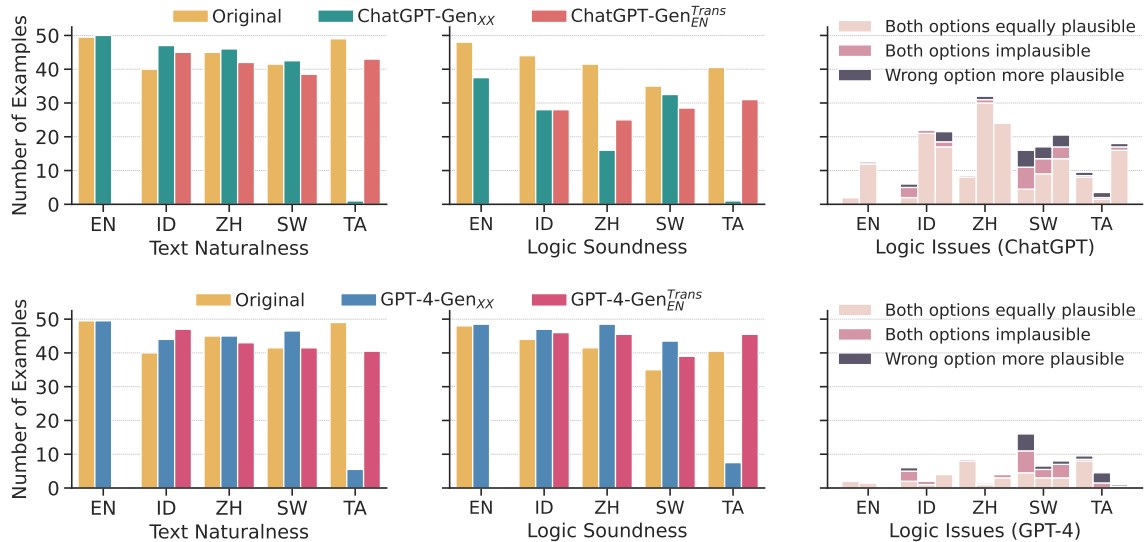

Figure 1: Human evaluation of 50 random examples from the original XCOPA, ChatGPT (top) and GPT-4 (bottom) generated data in target languages, and translation of English generated data. Examples are annotated by two native speakers in each language. The subplots in the last column show the logic issues of the XCOPA data, where the three bars for each language represent $Oringal$, $Gen_{XX}$, and $Gen_{EN}^{Trans}$ (from left to right).

the 3.6k dataset is around 360, it is plausible that training solely on those examples could raise the accuracy level, or even surpass, what we observe for the entire dataset.[9] An intriguing question that remains to be investigated in future research is why the remaining 3.2k incorrect or unnatural examples do not negatively impact the model's performance.

The translated text is typically less natural than the original and generated data (apart from ID due to issues in the original data). This result affirms that LLMs generally excel in generating fluent text for the languages it supports.

### 6.2 Logic Soundness

In terms of logic soundness, ChatGPT falls short compared to the original dataset. We further illustrate the categorised issues in the last column of the plots in Figure 1. We can see that for ChatGPT, the majority of the examples are labelled as "both options are equally plausible", only SW has more problematic examples with "the wrong option is actually more plausible". We suspect that this issue arises from the instruction provided (taken from the description of the original COPA dataset), which states that "both options could be plausible, but one is more plausible." In some cases, ChatGPT generates two choices that are excessively similar in terms of plausibility. On the other hand, GPT-4

tends to generate options with more clear-cut differences in plausibility, mirroring the original data. We note that despite the description/instruction that both alternatives could happen, both the original dataset and the data synthesised by GPT-4 tend to present one plausible and one *implausible* option.

For English XWinograd and XstoryCloze, the majority of the examples in both original and generated examples are evaluated as natural and logically sound. For XWinograd, although more than 47 examples are evaluated to exhibit high text quality and follow commonsense logic, only 23 ChatGPT-generated examples fulfil the requirement that both noun phrases should be interchangeable with the same pronoun. GPT-4 examples demonstrate better consistency, with 36 following this rule, whereas all original examples are found satisfactory.

### 7 Conclusions

This paper explores the effectiveness of utilising LLMs for data augmentation in cross-lingual datasets with limited training data. We specifically focus on commonsense reasoning tasks that are challenging for data synthesis. Our experiments including four LLMs for data generation on three datasets, showcase enhanced cross-lingual zero-shot transfer on smaller fine-tuned task-specific language models. However, the impact varies across different datasets and languages. Notably, larger models such as XLMR-Large, which have higher

---

[9]We could not conduct this experiment as the entire dataset was not manually labelled.

baselines, demonstrate more difficulty in achieving performance improvements with LLM-generated data. Among the four LLMs, GPT-4-generated data exhibits mostly consistent superior performance.

Expanding data generation directly in target languages also shows general improvements compared to cross-lingual zero-shot with the English-generated data. Human evaluation of the synthesised multilingual dataset shows that the ChatGPT and GPT-4 generated data demonstrate high naturalness in most languages, even surpassing the original data. However, in certain languages like TA, both models fail to generate natural text. Additionally, when assessing the logical soundness of the dataset, examples synthesised by ChatGPT reveal notable inconsistencies regarding more plausible options compared to the original human-created data. In contrast, GPT-4 exhibits a performance on par with human-written data.

In conclusion, leveraging LLMs for data augmentation shows promise. However, the choice of LLM used for data generation significantly influences the quality of the resulting data, as well as its applicability to the language under consideration. In circumstances where a more advanced model such as GPT-4 cannot be accessed, other models can be utilised, though this might result in performance difficulties in certain non-English languages - a challenge that also exists for GPT-4 - and concerns regarding logical coherence. A compelling direction for future research could involve exploring the efficacy of more recent instruction-tuned or aligned open-source LLMs, such as LLaMA 2 (Touvron et al., 2023b) or TÜLU (Wu et al., 2023), in enhancing data augmentation.

## Limitations

We have identified the following limitations in this work: (1)While LLMs, especially GPT-4, exhibit promising results in the context of multilingual commonsense data augmentation, they may encounter challenges when applied to extremely low-resource languages. (2) In order to achieve optimal performance, few-shot examples in the target language are still necessary for generating new examples. However, acquiring such examples may not always be feasible for all languages of interest. (3) The usage of closed models like GPT-4 is limited by licensing restrictions, and the results obtained from these models may not be reproducible. Nonetheless, the experiments conducted in this study demonstrate the potential benefits of leveraging LLMs for multilingual dataset augmentation.

## Ethical Consideration

Synthetic data generation with LLMs, especially multilingual data, should be approached with sensitivity and respect, as it reflects the linguistic, social, and cultural identity of a multilingual community. Since LLMs are trained on web data, they may encode biases perpetuating stereotypes, discrimination, or marginalisation of specific languages or communities. Therefore, collaboration with linguists, language experts, and community representatives is necessary to avoid the unintentional perpetuation of stereotypes and cultural insensitivity.

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

## A  Model Details

The open-source models used in the experiments are as follows:

- mBERT: `https://huggingface.co/bert-base-multilingual-uncased`

- XLMR-base: `https://huggingface.co/xlm-roberta-base`

- XLMR-large: `https://huggingface.co/xlm-roberta-large`

- Dolly-v2: `https://huggingface.co/databricks/dolly-v2-12b`

- StableVinuca: `https://huggingface.co/CarperAI/stable-vicuna-13b-delta`

## B  Sentences and Event Diversity of ChatGPT-generated StoryCloze Data

As the StoryCloze dataset contains more sentences and has richer content, we follow the analysis of the ROC story and further compare the stylistic features in terms of sentence length, and the most frequent events[10] generated by ChatGPT with the original data. This helps us to determine whether ChatGPT-generated data can capture the corpus distribution by randomly sampling $n$ examples from the dataset in the instructions.

In Figure 2, we present the results of comparing the generated data points with the original 300 train set used as few-shot examples in the generation instructions. We can see that 23 of the 30 most frequent events in the original dataset can also be found in the 30 most frequent events of the ChatGPT-generated data. Regarding the sentence length, we observe that ChatGPT tends to

---

[10]Here we follow Mostafazadeh et al. (2016) where an event is counted as any hyponym of "event" or "process" in WordNet.

| Model | Ratio | XCOPA | XWingrad | XStoryCloze |
|---|---|---|---|---|
| mBERT | 1× | 64.0 | 50.2 | 74.6 |
| | 2× | 64.8 | 51.9 | 76.8 |
| | 5× | 68.0 | 57.1 | 80.6 |
| | 10× | 69.8 | 65.7 | 80.3 |
| XLMR-Base | 1× | 58.0 | 45.9 | 70.7 |
| | 2× | 59.0 | 53.7 | 79.7 |
| | 5× | 63.0 | 67.8 | 81.9 |
| | 10× | 65.8 | 71.2 | 84.1 |
| XLMR-Large | 1× | 56.0 | 78.1 | 81.1 |
| | 2× | 61.2 | 79.8 | 90.9 |
| | 5× | 81.4 | 82.0 | 89.9 |
| | 10× | 85.2 | 82.8 | 91.9 |

Table 7: Performance on English test examples training on GPT-4-generated English data and the original data. Original data points selected from the three datasets are set to 200. 1× corresponds to using only the original data, 2× means using 200 original data and 200 generated data.

generate longer sentences, especially for the ending sentences, whereas in the original dataset, they tend to be the shortest among all sentences.

## C  Fixed Ratio Data Augmentation

We experiment with generating data with a fixed ratio of the original datasets. Specifically, we compare training with the original English data (200 randomly selected examples) and augment it with different quantities of English examples generated by GPT-4, where we include original training instances in all cases.

The results in Table 7 showcase the performance on English test examples when fine-tuning mBERT and XLMR models with training data sizes that are 1×, 2×, 5×, and 10× the size of the original dataset. We can see that performance consistently improves as we increase the amount of generated data except XStoryCloze, which has the highest baselines, echoing the previous findings. The relative performance gain is generally more pronounced when increasing the data from 2× to 5× for the other two datasets.

## D  Additional Results

This section includes the following additional results: Table 8, Table 9, and Table 10 show generated data in English with different LLMs on XCOPA, XWinograd, and XStoryCloze. Table 11 and Table 12 show the full result on XCOPA with ChatGPT and GPT-4.

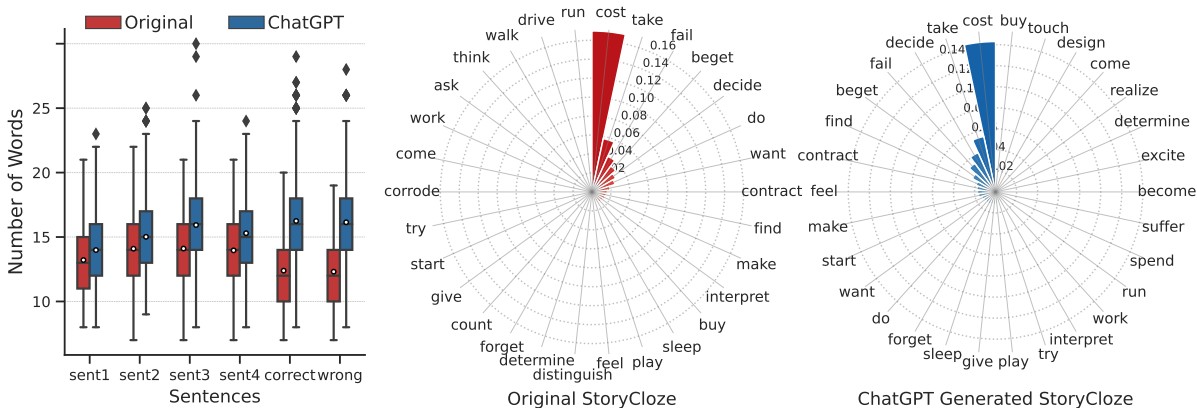

Figure 2: Comparison between the 30 most frequent events and the lengths of the sentences in the original and the ChatGPT-generated English StoryCloze dataset.

| Fine-tuned | Train Data | LLM | AVG | EN | ET | HT | ID | IT | QU | SW | TA | TH | TR | VI | ZH |
|---|---|---|---|---|---|---|---|---|---|---|---|---|---|---|---|
| MBERT | GEN | Dolly-v2 | 54.0 | 63.4 | 52.0 | 52.2 | 54.0 | 53.8 | 47.6 | 48.6 | 53.4 | **53.4** | 52.8 | 50.4 | 58.2 |
| | | StableVicuna | 53.5 | 62.4 | 51.6 | 49.2 | 55.8 | 55.8 | 50.0 | **50.2** | 50.2 | 52.6 | 51.0 | 50.4 | 56.0 |
| | | ChatGPT | 56.0 | 64.8 | 54.8 | 52.6 | 58.0 | 57.4 | 49.8 | 48.4 | **55.6** | 52.8 | 53.2 | 53.0 | 59.0 |
| | | GPT-4 | **58.2** | 69.2 | 59.2 | 54.0 | 60.6 | 59.2 | 50.8 | 48.2 | 55.0 | 48.2 | 53.8 | 57.6 | 61.0 |
| | GEN+ORI | Dolly-v2 | 54.4 | 59.8 | 52.6 | 53.2 | 53.0 | 56.4 | 53.8 | 52.4 | 50.4 | 54.8 | 49.8 | 52.6 | 58.8 |
| | | StableVicuna | 55.6 | 65.2 | 53.4 | 50.4 | 59.0 | 60.0 | 51.6 | 50.4 | 49.4 | 52.0 | 52.4 | 54.0 | 58.2 |
| | | ChatGPT | 54.6 | 59.6 | 56.4 | 53.6 | 53.8 | 51.4 | 51.4 | 51.6 | 50.4 | 52.6 | 54.0 | 55.0 | 59.2 |
| | | GPT-4 | **59.3** | 72.6 | 58.8 | 53.0 | 62.0 | 61.0 | 53.0 | 50.0 | 54.0 | 48.2 | 52.0 | 57.6 | 64.6 |
| XLMR-Base | GEN | Dolly-v2 | 59.0 | 64.4 | 58.8 | 52.8 | 60.8 | 61.0 | 50.8 | 55.6 | 60.4 | 58.0 | 57.2 | 58.6 | 59.0 |
| | | StableVicuna | 58.5 | 60.4 | 59.4 | **53.6** | 60.8 | 56.8 | 49.2 | 56.0 | 61.2 | 60.4 | 54.8 | 59.6 | 58.6 |
| | | ChatGPT | 58.8 | 62.4 | 56.4 | 52.4 | 61.4 | 58.6 | **52.2** | 52.0 | 63.4 | 61.2 | 56.4 | 59.6 | 62.8 |
| | | GPT-4 | **63.6** | 67.0 | 62.4 | 52.0 | 68.6 | 62.6 | 51.8 | **58.6** | 65.4 | 64.8 | 63.2 | 66.6 | 69.6 |
| | GEN+ORI | Dolly-v2 | 58.7 | 65.6 | 57.6 | **52.2** | 60.8 | 58.4 | 52.4 | 58.2 | 57.4 | 58.0 | 58.4 | 58.0 | 59.8 |
| | | StableVicuna | 61.1 | 65.0 | 62.4 | 49.4 | 64.2 | 62.4 | 46.2 | **60.4** | 59.6 | 58.0 | 58.0 | 63.0 | 63.4 |
| | | ChatGPT | 59.8 | 63.8 | 61.6 | 51.6 | 62.6 | 59.8 | 51.2 | 51.6 | 60.4 | 61.6 | 61.8 | 64.8 | 62.0 |
| | | GPT-4 | **63.6** | 69.6 | 63.8 | 51.2 | 67.2 | 62.4 | 52.6 | 58.4 | 63.8 | 66.0 | 64.2 | 66.8 | 69.4 |
| XLMR-Large | GEN | Dolly-v2 | 59.6 | 62.4 | 58.6 | 49.6 | 64.8 | 59.2 | 50.6 | 56.8 | 60.8 | 58.8 | 57.0 | 61.0 | 63.0 |
| | | StableVicuna | 65.7 | 71.4 | 66.2 | 50.4 | 71.4 | 70.2 | 50.0 | 60.0 | 64.0 | 63.6 | 68.0 | 68.2 | 69.8 |
| | | ChatGPT | 65.2 | 71.2 | 64.6 | 51.6 | 70.8 | 66.6 | **51.0** | 58.8 | 66.0 | 68.2 | 69.0 | 68.8 | 68.8 |
| | | GPT-4 | **73.6** | 83.2 | 71.2 | 52.0 | 81.2 | 78.2 | 51.0 | 62.2 | 76.6 | 77.4 | 75.0 | 78.4 | 79.0 |
| | GEN+ORI | Dolly-v2 | 66.4 | 74.2 | 62.8 | **53.0** | 72.0 | 70.4 | 46.2 | 61.6 | 65.6 | 66.2 | 69.6 | 67.6 | 70.6 |
| | | StableVicuna | 69.9 | 76.0 | 69.8 | 51.2 | 75.0 | 74.2 | 51.2 | 64.4 | 70.2 | 71.6 | 72.2 | 72.6 | 75.4 |
| | | ChatGPT | 69.5 | 76.4 | 69.8 | 48.2 | 76.0 | 72.8 | 50.8 | 63.4 | 67.8 | 70.8 | 70.2 | 73.4 | 77.8 |
| | | GPT-4 | **73.7** | 84.6 | 70.4 | 50.0 | 80.8 | 80.2 | 51.8 | 65.8 | 72.8 | 76.0 | 74.8 | 78.4 | 80.4 |

Table 8: Accuracy on XCOPA with English generated data from different LLMs.

| Fine-tuned | Training data | LLM | AVG | EN | FR | JA | PT | RU | ZH |
|---|---|---|---|---|---|---|---|---|---|
| MBERT | GEN | Dolly-v2 | **56.47** | **71.24** | 53.01 | 52.45 | **53.23** | 54.92 | 53.97 |
| | | StableVicuna | 53.73 | 54.94 | **56.63** | 50.26 | 50.57 | 52.06 | 57.94 |
| | | ChatGPT | 56.00 | 54.94 | 54.22 | **54.01** | 52.09 | **55.87** | **64.88** |
| | | GPT-4 | 54.90 | 56.22 | **56.63** | 52.55 | 51.71 | 52.38 | 59.92 |
| | GEN+ORI | Dolly-v2 | **59.32** | **71.24** | 57.83 | **53.81** | 56.65 | 59.05 | 57.34 |
| | | StableVicuna | 58.46 | 57.94 | 63.86 | **53.81** | **57.41** | 58.41 | 59.33 |
| | | ChatGPT | 58.26 | 56.65 | **66.27** | 53.60 | 56.27 | **60.00** | 56.75 |
| | | GPT-4 | 57.48 | 53.65 | 62.65 | **54.43** | 55.89 | 57.14 | **61.11** |
| XLMR-Base | GEN | Dolly-v2 | 59.63 | **71.24** | 57.83 | 55.79 | **57.03** | 57.78 | 58.13 |
| | | StableVicuna | 58.95 | 60.09 | 55.42 | 57.35 | 52.47 | 58.73 | 69.64 |
| | | ChatGPT | 62.69 | 69.10 | 60.24 | 61.42 | **57.03** | 61.27 | 67.06 |
| | | GPT-4 | **63.32** | 69.10 | **61.45** | 61.52 | 56.65 | 60.95 | 70.24 |
| | GEN+ORI | Dolly-v2 | 66.33 | **75.54** | 63.86 | 65.80 | **64.26** | 62.86 | 65.67 |
| | | StableVicuna | 65.97 | 64.38 | 66.27 | 67.15 | 63.88 | **65.71** | 68.45 |
| | | ChatGPT | 65.94 | 65.24 | 60.24 | **68.93** | 70.72 | 62.86 | 67.66 |
| XLMR-Large | GEN | GPT-4 | **66.88** | 68.24 | **67.47** | 66.94 | 63.88 | 63.49 | **71.23** |
| | | Dolly-v2 | **76.86** | **87.55** | 67.47 | **81.02** | **76.43** | 74.29 | 74.40 |
| | | StableVicuna | 68.22 | 74.25 | 63.86 | 68.20 | 66.16 | 63.81 | 73.02 |
| | | ChatGPT | 73.20 | 81.97 | 66.27 | 73.10 | 66.92 | 72.38 | 78.57 |
| | | GPT-4 | 76.37 | 81.55 | **74.70** | 75.91 | 71.86 | **75.24** | 78.97 |
| | GEN+ORI | Dolly-v2 | 83.10 | **90.56** | 79.52 | 85.19 | 84.03 | 80.95 | 78.37 |
| | | StableVicuna | 82.02 | 83.26 | 80.72 | 83.84 | 86.31 | **82.22** | 75.79 |
| | | ChatGPT | 83.22 | 85.84 | 80.72 | **87.38** | 85.93 | 80.95 | 78.50 |
| | | GPT-4 | **83.52** | 85.41 | **81.93** | 85.92 | **86.69** | 80.63 | **80.56** |

Table 9: Accuracy on XWinograd with English generated data from different LLMs.

| Fine-tuned | Training data | LLM | AVG | EN | RU | ZH | ES | AR | HI | ID | TE | SW | EU | MY |
|---|---|---|---|---|---|---|---|---|---|---|---|---|---|---|
| MBERT | GEN | Dolly-v2 | **68.7** | **78.8** | **71.3** | **73.6** | **74.2** | 67.4 | 66.9 | 69.0 | **65.0** | 60.9 | **66.8** | 62.0 |
| | | StableVicuna | 64.6 | 71.4 | 66.8 | 68.8 | 68.1 | 64.3 | 63.6 | 66.1 | 61.2 | 58.6 | 63.6 | 58.4 |
| | | ChatGPT | 64.3 | 69.7 | 66.4 | 68.1 | 68.0 | 64.6 | 64.5 | 66.6 | 59.8 | 59.2 | 62.3 | 58.4 |
| | | GPT-4 | 68.0 | 75.5 | 70.8 | 73.3 | 70.4 | **67.6** | **68.2** | **69.6** | 63.1 | **62.3** | 65.4 | **62.2** |
| | GEN+ORI | Dolly-v2 | 68.1 | 75.7 | 71.2 | 72.4 | 73.2 | 66.4 | 67.1 | 68.9 | **64.5** | 61.4 | 67.1 | 61.0 |
| | | StableVicuna | 67.3 | 77.0 | 71.0 | 70.2 | 71.4 | 67.2 | 66.5 | 68.4 | 62.4 | 60.5 | 64.3 | 61.4 |
| | | ChatGPT | 68.3 | 76.4 | 68.5 | 72.9 | 73.0 | 66.3 | 68.6 | 71.1 | 62.0 | **62.0** | 67.4 | 63.4 |
| | | GPT-4 | **69.8** | **79.5** | **73.1** | **75.3** | 73.4 | **68.1** | **69.8** | 71.9 | 64.1 | **62.0** | **68.9** | 61.6 |
| XLMR-Base | GEN | Dolly-v2 | 75.8 | 81.4 | 79.2 | 80.3 | 78.0 | 73.6 | 74.7 | 80.7 | 73.0 | 68.8 | 72.2 | 71.7 |
| | | StableVicuna | 69.6 | 72.3 | 71.1 | 71.5 | 70.4 | 68.3 | 70.4 | 72.1 | 68.4 | 65.7 | 68.0 | 67.7 |
| | | ChatGPT | 67.4 | 69.7 | 68.9 | 68.5 | 68.7 | 66.1 | 68.2 | 68.7 | 67.0 | 63.7 | 65.6 | 66.6 |
| | | GPT-4 | **74.6** | 78.2 | 78.0 | 78.1 | 77.0 | 73.5 | 75.7 | 77.6 | 71.7 | 68.4 | 73.6 | 69.2 |
| | GEN+ORI | Dolly-v2 | 76.5 | 81.5 | 80.0 | 80.5 | 79.4 | 75.1 | 75.0 | 79.6 | 74.5 | 71.5 | 72.3 | 72.6 |
| | | StableVicuna | 74.2 | 79.2 | 77.4 | 77.8 | 76.4 | 74.0 | 74.5 | 78.2 | 70.2 | 67.6 | 71.7 | 69.6 |
| | | ChatGPT | 74.5 | 78.0 | 76.6 | 78.8 | 76.2 | 72.9 | 73.9 | 78.9 | 71.5 | 69.6 | 72.3 | 71.0 |
| | | GPT-4 | **79.3** | **85.4** | **83.2** | **82.6** | **83.0** | **78.0** | **79.9** | **82.7** | **75.9** | **72.9** | **74.9** | **74.3** |
| XLMR-Large | GEN | Dolly-v2 | 84.8 | 87.4 | 87.3 | 87.8 | 86.6 | 83.0 | 84.4 | 87.1 | **84.1** | 81.0 | 82.9 | 81.4 |
| | | StableVicuna | 74.6 | 76.7 | 75.9 | 77.4 | 76.2 | 72.9 | 74.5 | 76.2 | 74.3 | 70.8 | 73.5 | 72.5 |
| | | ChatGPT | 77.3 | 78.6 | 79.9 | 78.0 | 77.9 | 75.8 | 77.4 | 78.0 | 76.4 | 73.5 | 77.1 | 77.7 |
| | | GPT-4 | **86.0** | **88.5** | **88.2** | **88.2** | **88.0** | **84.9** | **85.7** | **87.8** | 83.7 | **81.3** | **85.6** | **84.3** |
| | GEN+ORI | Dolly-v2 | 86.4 | 89.2 | 87.2 | 89.5 | 87.1 | 85.2 | 86.7 | 87.7 | **85.0** | 83.0 | 85.7 | 83.8 |
| | | StableVicuna | 84.8 | 88.4 | 87.6 | 87.8 | 86.6 | 82.9 | 83.3 | 87.4 | 83.7 | 81.3 | 83.7 | 80.0 |
| | | ChatGPT | 85.8 | 88.5 | 88.0 | 88.3 | 87.3 | 83.7 | 85.9 | 87.2 | 83.7 | 81.6 | 85.4 | 83.8 |
| | | GPT-4 | **88.4** | **92.3** | **91.5** | **91.5** | **90.5** | **86.4** | **88.4** | **91.1** | 84.8 | **83.1** | **87.4** | **85.2** |

Table 10: Accuracy on XStoryCloze with English generated data from different LLMs.

| Model | Training Data | \|Data\| | AVG | EN | ET | HT | ID | IT | QU | SW | TA | TH | TR | VI | ZH |
|---|---|---|---|---|---|---|---|---|---|---|---|---|---|---|---|
| MBERT | *ORI* (BASELINE) | 400 | 47.2 | 53.8 | 44.2 | 48.6 | 47.2 | 46.2 | 50.6 | 45.4 | 48.4 | 49.8 | 49.8 | 43.6 | 47.4 |
| | $GEN_{EN}$ | 3.7k | 56.0 | 64.8 | 54.8 | 52.6 | 58.0 | 57.4 | 49.8 | 48.4 | 55.6 | 52.8 | 53.2 | 53.0 | 59.0 |
| | $GEN_{EN}$ + *ORI* | 4.1k | 54.6 | 59.6 | 56.4 | 53.6 | 53.8 | 51.4 | 51.4 | 51.6 | 50.4 | 52.6 | 54.0 | 55.0 | 59.2 |
| | $GEN_{EN}$ + *ORI* (TLV) | 4.1k | 57.6 | 68.0 | 55.4 | 54.0 | 61.2 | 59.8 | 51.8 | 51.2 | 55.8 | 54.4 | 52.2 | 53.4 | 59.2 |
| | $GEN_{EN}$ | 28.6k | 57.2 | 66.2 | 55.8 | 50.8 | 58.6 | 58.2 | 53.2 | 51.2 | 57.2 | 53.2 | 52.0 | 56.0 | 61.0 |
| | $GEN_{EN}$ + *ORI* | 29k | 57.0 | 66.6 | 55.4 | 51.4 | 59.2 | 58.6 | 52.4 | 50.8 | 53.6 | 53.2 | 50.0 | 54.8 | 62.8 |
| | $GEN_{EN}$ + *ORI* (TLV) | 29k | 57.0 | 66.6 | 55.4 | 51.4 | 59.2 | 58.6 | 52.4 | 50.8 | 53.6 | 53.2 | 50.0 | 54.8 | 62.8 |
| | $GEN_{XX}$ | 3.6k/lang | 57.5 | 64.8 | 57.8 | 57.4 | 58.0 | 60.2 | 54.6 | 51.4 | 53.0 | – | – | 53.0 | 62.0 |
| | $GEN_{XX}$ + *ORI* | 4k | 56.8 | 59.6 | 58.8 | 54.6 | 56.2 | 61.2 | 53.6 | 54.6 | 53.6 | – | – | 52.0 | 60.2 |
| | $GEN_{EN}^{Trans}$ + *ORI* | 4k | 58.7 | 59.6 | 59.8 | 59.8 | 62.8 | 61.0 | – | 52.6 | 56.8 | 53.4 | 56.2 | 58.2 | 59.4 |
| | $GEN_{EN}^{Trans}$ + *ORI* | 29k/lang | **60.6** | 66.6 | 61.8 | 57.8 | 60.8 | 62.2 | – | 53.2 | 58.4 | 53.2 | 63.0 | 60.6 | 63.8 |
| XLMR-BASE | *ORI* (BASELINE) | 400 | 55.6 | 57.6 | 54.6 | 50.6 | 59.6 | 54.8 | 46.0 | 55.0 | 53.4 | 56.2 | 55.2 | 54.8 | 59.6 |
| | $GEN_{EN}$ | 3.7k | 58.8 | 62.4 | 56.4 | 52.4 | 61.4 | 58.6 | 52.2 | 52.0 | 63.4 | 61.2 | 56.4 | 59.6 | 62.8 |
| | $GEN_{EN}$ + *ORI* | 4.1k | 59.8 | 63.8 | 61.6 | 51.6 | 62.6 | 59.8 | 51.2 | 51.6 | 60.4 | 61.6 | 61.8 | 64.8 | 62.0 |
| | $GEN_{EN}$ + *ORI* (TLV) | 4.1k | 60.7 | 63.2 | 61.6 | 51.4 | 64.8 | 61.2 | 51.2 | 53.6 | 62.6 | 63.0 | 58.2 | 61.0 | 66.6 |
| | $GEN_{EN}$ | 28.6k | 60.8 | 66.4 | 57.2 | 56.0 | 66.4 | 61.2 | 53.0 | 53.8 | 60.0 | 61.6 | 56.6 | 61.4 | 64.6 |
| | $GEN_{EN}$ + *ORI* | 29k | 62.1 | 64.6 | 61.8 | 50.6 | 66.8 | 63.6 | 48.0 | 55.6 | 65.8 | 63.6 | 57.2 | 63.2 | 66.8 |
| | $GEN_{EN}$ + *ORI* (TLV) | 29k | 60.9 | 66.4 | 61.8 | 49.8 | 66.2 | 59.8 | 54.6 | 53.4 | 62.4 | 63.8 | 58.2 | 62.8 | 65.8 |
| | $GEN_{XX}$ | 3.6k/lang | 58.8 | 62.4 | 57.0 | 55.6 | 61.4 | 59.0 | 55.6 | 54.4 | 56.8 | – | – | 60.6 | 62.0 |
| | $GEN_{XX}$ + *ORI* | 4k | 59.9 | 63.8 | 60.6 | 55.0 | 64.6 | 59.6 | 52.6 | 54.6 | 56.4 | – | – | 59.6 | 64.8 |
| | $GEN_{EN}^{Trans}$ + *ORI* | 4k | 61.1 | 63.8 | 60.0 | 58.0 | 65.0 | 60.8 | – | 53.8 | 60.2 | 66.2 | 56.6 | 62.6 | 66.0 |
| | $GEN_{EN}^{Trans}$ + *ORI* | 29k/lang | **62.2** | 64.6 | 63.2 | 57.2 | 64.8 | 61.2 | – | 55.0 | 61.2 | 59.2 | 59.5 | 64.2 | 68.4 |
| XLMR-LARGE | *ORI* (BASELINE) | 400 | 64.4 | 71.4 | 62.8 | 51.4 | 69.0 | 65.8 | 52.0 | 60.6 | 62.0 | 64.0 | 61.2 | 69.4 | 66.8 |
| | $GEN_{EN}$ | 3.7k | 65.2 | 71.2 | 64.6 | 51.6 | 70.8 | 66.6 | 51.0 | 58.8 | 66.0 | 68.2 | 69.0 | 68.8 | 68.8 |
| | $GEN_{EN}$ + *ORI* | 4.1k | 69.5 | 76.4 | 69.8 | 48.2 | 76.0 | 72.8 | 50.8 | 63.4 | 67.8 | 70.8 | 70.2 | 73.4 | 77.8 |
| | $GEN_{EN}$ + *ORI* (TLV) | 4.1k | 71.9 | 80.6 | 71.6 | 50.8 | 78.6 | 77.2 | 51.8 | 63.0 | 69.2 | 71.2 | 72.8 | 77.2 | 78.8 |
| | $GEN_{EN}$ | 28.6k | 71.8 | 80.6 | 74.4 | 51.0 | 78.4 | 75.2 | 51.2 | 63.4 | 69.8 | 70.6 | 69.8 | 75.6 | 77.4 |
| | $GEN_{EN}$ + *ORI* | 29k | **72.4** | 81.0 | 73.8 | 54.4 | 80.2 | 75.2 | 48.8 | 61.4 | 70.4 | 73.8 | 70.4 | 75.6 | 79.8 |
| | $GEN_{EN}$ + *ORI* (TLV) | 29k | **72.4** | 81.0 | 73.8 | 54.4 | 80.2 | 75.2 | 48.8 | 61.0 | 70.4 | 73.8 | 70.4 | 75.6 | 79.8 |
| | $GEN_{XX}$ | 3.6k/lang | 63.4 | 71.2 | 62.6 | 54.2 | 71.0 | 65.8 | 49.4 | 53.8 | 56.4 | – | – | 64.0 | 71.6 |
| | $GEN_{XX}$ + *ORI* | 4k | 65.2 | 76.4 | 62.4 | 55.2 | 75.0 | 62.2 | 54.0 | 58.2 | 55.4 | – | – | 66.2 | 76.2 |
| | $GEN_{EN}^{Trans}$ + *ORI* | 4k | 67.0 | 76.4 | 60.0 | 59.6 | 66.2 | 66.6 | – | 59.0 | 64.8 | 71.2 | 65.2 | 74.8 | 75.6 |
| | $GEN_{EN}^{Trans}$ + *ORI* | 29k/lang | 71.5 | 81.0 | 71.8 | 57.2 | 79.8 | 74.4 | – | 54.8 | 71.4 | 72.6 | 70.0 | 77.2 | 75.6 |

Table 11: Full results on XCOPA (with ChatGPT-generated data). +TLV corresponds to including the original validation set in all Target Languages in the Validation set. Rows are sorted by the number of instances used in training. AVG shows average results for languages that are available in all settings (excl. QU, TH, TR).

| Model | Training Data | AVG | EN | ET | HT | ID | IT | QU | SW | TA | TH | TR | VI | ZH |
|---|---|---|---|---|---|---|---|---|---|---|---|---|---|---|
| mBERT | $ORI$ | 47.2 | 53.8 | 44.2 | 48.6 | 47.2 | 46.2 | 50.6 | 45.4 | 48.4 | 49.8 | 49.8 | 43.6 | 47.4 |
| | $GEN_{EN}$ | 58.2 | 69.2 | 59.2 | 54.0 | 60.6 | 59.2 | 50.8 | 48.2 | 55.0 | 48.2 | 53.8 | 57.6 | 61.0 |
| | $GEN_{EN} + ORI$ | 59.3 | 72.6 | 58.8 | 53.0 | 62.0 | 61.0 | 53.0 | 50.0 | 54.0 | 48.2 | 52.0 | 57.6 | 64.6 |
| | $GEN_{XX}$ | 60.2 | 69.2 | 59.4 | 56.2 | 60.2 | 63.8 | 54.4 | 55.2 | 54.0 | – | – | 61.2 | 62.2 |
| | $GEN_{XX} + ORI$ | 61.8 | 72.6 | 61.2 | 58.2 | 62.2 | 66.4 | 54.4 | 57.4 | 53.4 | – | – | 63.0 | 61.8 |
| | $GEN_{EN}^{Trans}$ | 61.4 | 69.2 | 59.2 | 56.8 | 65.4 | 65.2 | – | 53.4 | 56.8 | 52.6 | 59.6 | 61.8 | 65.0 |
| | $GEN_{EN}^{Trans} + ORI$ | 62.6 | 72.6 | 58.6 | 55.2 | 65.6 | 65.4 | – | 53.8 | 62.6 | 53.2 | 58.8 | 64.6 | 65.4 |
| XLMR-Base | $ORI$ | 55.6 | 57.6 | 54.6 | 50.6 | 59.6 | 54.8 | 46.0 | 55.0 | 53.4 | 56.2 | 55.2 | 54.8 | 59.6 |
| | $GEN_{EN}$ | 63.6 | 67.0 | 62.4 | 52.0 | 68.6 | 62.6 | 51.8 | 58.6 | 65.4 | 64.8 | 63.2 | 66.6 | 69.6 |
| | $GEN_{EN} + ORI$ | 63.6 | 69.6 | 63.8 | 51.2 | 67.2 | 62.4 | 52.6 | 58.4 | 63.8 | 66.0 | 64.2 | 66.8 | 69.4 |
| | $GEN_{XX}$ | 63.2 | 67.0 | 60.8 | 56.4 | 68.6 | 62.4 | 57.4 | 58.2 | 60.2 | – | – | 64.6 | 70.4 |
| | $GEN_{XX} + ORI$ | 64.0 | 69.6 | 62.2 | 56.2 | 68.6 | 63.8 | 56.8 | 57.8 | 61.2 | – | – | 66.8 | 70.0 |
| | $GEN_{EN}^{Trans}$ | 62.5 | 67.0 | 60.0 | 55.6 | 66.0 | 62.4 | – | 58.0 | 60.4 | 64.4 | 64.6 | 64.0 | 68.8 |
| | $GEN_{EN}^{Trans} + ORI$ | 63.9 | 69.6 | 61.6 | 56.6 | 68.4 | 65.2 | – | 58.2 | 60.2 | 68.0 | 62.6 | 66.0 | 69.6 |
| XLMR-Large | $ORI$ | 64.4 | 71.4 | 62.8 | 51.4 | 69.0 | 65.8 | 52.0 | 60.6 | 62.0 | 64.0 | 61.2 | 69.4 | 66.8 |
| | $GEN_{EN}$ | 73.6 | 83.2 | 71.2 | 52.0 | 81.2 | 78.2 | 51.0 | 62.2 | 76.6 | 77.4 | 75.0 | 78.4 | 79.0 |
| | $GEN_{EN} + ORI$ | 73.7 | 84.6 | 70.4 | 50.0 | 80.8 | 80.2 | 51.8 | 65.8 | 72.8 | 76.0 | 74.8 | 78.4 | 80.4 |
| | $GEN_{XX}$ | 72.8 | 83.2 | 75.2 | 55.2 | 78.4 | 76.0 | 52.4 | 63.0 | 68.2 | – | – | 77.8 | 78.6 |
| | $GEN_{XX} + ORI$ | 74.6 | 84.6 | 77.0 | 56.0 | 82.2 | 77.0 | 56.0 | 65.0 | 73.8 | – | – | 76.2 | 80.0 |
| | $GEN_{EN}^{Trans}$ | 71.0 | 83.2 | 72.4 | 55.6 | 79.4 | 78.2 | – | 60.6 | 67.8 | 77.8 | 72.6 | 64.0 | 77.4 |
| | $GEN_{EN}^{Trans} + ORI$ | 74.1 | 84.6 | 74.2 | 57.2 | 82.0 | 77.4 | – | 62.2 | 75.0 | 75.2 | 72.8 | 74.4 | 79.6 |

Table 12: Accuracy on XCOPA. $GEN_{EN}$ and $GEN_{XX}$ represents 3.7K and 3.6K data in English and target languages generated by GPT-4. AVG shows average results for languages that are available in all settings (excl. QU, TH, TR).