# OpenReview forum: "LLM-powered Data Augmentation for Enhanced Cross-lingual Performance"
_EMNLP/2023/Conference — EMNLP 2023 Main_

### Official Review · Reviewer_7m4W · 2023-07-28

**Soundness:** 4

**Excitement:**

3: Ambivalent: It has merits (e.g., it reports state-of-the-art results, the idea is nice), but there are key weaknesses (e.g., it describes incremental work), and it can significantly benefit from another round of revision. However, I won't object to accepting it if my co-reviewers champion it.

**Paper Topic And Main Contributions:**

This paper explores data augmentation methods for multilingual commonsense reasoning tasks. The key idea in this work is to leverage few-shot prompting methods on large language models (like ChatGPT / GPT4) to generate a lot of task-specific English data. Smaller multilingual models like mBERT are then fine-tuned and evaluated zero-shot on other languages ("zero-shot crosslingual generalization").

Across three datasets, three multilingual models, and four data generating language models, the paper shows that the proposed data augmentation method is quite effective at improving zero-shot crosslingual generalization. The paper also conducts extensive analysis on the quality of the generated data (via human studies), and explores data augmentation using multilingual data as well.


**Reasons To Accept:**


1. The paper focuses on an important research problem in NLP: multilingual commonsense reasoning. In the last few years, large language modeling research has primarily focused on English. It's important to see how the benefits from English large language models can be used in other lower resource languages, and this paper does a good job studying this research question on commonsense reasoning tasks.

2. This paper has a comprehensive experimental setup, and experiments with 3 commonsense reasoning datasets, three multilingual models, and experiments with data generated by 3 large language models. The paper has experiments generating English examples (zero-shot cross lingual generalization), translated examples, as well as examples in the language itself. The paper also has some good analysis with human evaluation experiments on the quality of the generated examples.

3. This paper is very well written, and easy to follow.

4. The authors extensively discuss and are upfront about the limitations of their work.


**Reasons To Reject:**


1. I am wondering what is the effect of these data augmentation methods when the base model is made larger / more powerful. The current experiments focus on fairly small multilingual models (mBERT / XLM), which are <500M parameters. It would be interesting to see how these results generalize to larger models like mT5 or BLOOM. This is important because it's possible that these larger models need lesser examples to generalize, and data augmentation will not as useful then.

2. A related point to weakness #1, the paper is missing baseline experiments which use the large language models themselves to perform the task, through few-shot learning or fine-tuning on original data using the GPT3 fine-tuning API (please me know in case I missed these experiments while reading the paper!). Since models like ChatGPT / GPT-4 are effectively used to create training data for downstream fine-tuning, I'm suspecting they can already perform the task pretty well?

3. This has been acknowledged by the authors in their limitations section, but there are several commerical restrictions on using ChatGPT/GPT4 outputs, [1] says "one may not use output from the Services to develop models that compete with OpenAI". This may restrict the applicability of the proposed approach for downstream applications. In future versions of the paper, it will be helpful to see how stronger instruction-tuned / aligned open LLMs like LLAMA 2 [2] or TULU [3] can help with the data augmentation. However, note that both these models came out in the last two months (during / after EMNLP submission period), so this is not a valid weakness for the current version of the paper.

4. Finally, in the long term I am not very convinced by using one language model

[1] - https://openai.com/policies/terms-of-use
[2] - https://huggingface.co/blog/llama2
[3] - https://arxiv.org/abs/2306.04751

**Reproducibility:**

4: Could mostly reproduce the results, but there may be some variation because of sample variance or minor variations in their interpretation of the protocol or method.

**Reviewer Confidence:**

4: Quite sure. I tried to check the important points carefully. It's unlikely, though conceivable, that I missed something that should affect my ratings.

---

> ### Author Rebuttal · Authors · 2023-08-28
>
> We appreciate the reviewers' detailed feedback and suggestions. We would like to address the concerns as follows:
>
> 1. One of the motivations and the main focus of this study is to evaluate the effectiveness of augmentation with synthetic data generated from large/better models on smaller models (line 10, 43). We believe the impact will be less profound in larger models as we can observe in XLMR-large, owing to their superior capacity and task baselines.
> 2. We acknowledge it is a great suggestion to report the performance of the LLMs used for data augmentation on the tasks, which will also provide an indication of the dataset quality. We will add this in the camera-ready version.
> 3. We appreciate the reviewer’s suggestion on future exploring directions and agree that studying the most recent open-source LLMs such as LLAMA 2 would provide great insights on data augmentation.
> 4. Our current work focuses on showcasing the potential of data augmentation from different LLMs, however, we agree that studying the mix of multiple models for data augmentation is a great direction to explore.

---

### Official Review · Reviewer_AwiM · 2023-07-29

**Soundness:** 3

**Excitement:**

3: Ambivalent: It has merits (e.g., it reports state-of-the-art results, the idea is nice), but there are key weaknesses (e.g., it describes incremental work), and it can significantly benefit from another round of revision. However, I won't object to accepting it if my co-reviewers champion it.

**Paper Topic And Main Contributions:**

The authors of this paper study the capabilities of large language models (LLMs) in performing data synthesis in cross-lingual, commonsense reasoning data sets. To this end, they leverage the in-context learning abilities of LLMs. Specifically, they construct data set-specific prompts that instruct the model to generate new data points based on a few given examples from the original data set. New data points are generated directly in the target language or English. In the latter case, they are translated to the target language using Google Translate API.

The authors employ four LLMs (Dolly-v2, StableVicuna, ChatGPT, and GPT-4) to augment three multilingual data sets (XCOPA, XWinograd, and XStoryCloze) and use the synthesized data to fine-tune two pre-trained multilingual transformer models (mBERT and XLMR). As a result, the models trained on augmented data sets exhibit improved accuracy. Overall, the best results are achieved with the data generated by the GPT-4 model. Moreover, training on an English data set translated to the target language performs generally better than using data generated directly in the target language. The authors also perform qualitative evaluation by asking native speakers to assess the generated examples in terms of naturalness and soundness.

In sum, this work makes the following contributions: 1) It proposes a straightforward LLM-based data synthesis method; 2) It presents the results of an experimental evaluation of the accuracy of two pre-trained multilingual transformer models fine-tuned with the synthesized data in a cross-lingual, commonsense reasoning task and 3) It performs a qualitative human evaluation of the examples generated by the proposed method.

**Questions For The Authors:**

Question A: What is the value of the "m" parameter mentioned in Table 2?

Question B: What value of the "temperature" hyper-parameter was used in the experiments?

**Reasons To Accept:**

The paper is well-motivated and studies an important problem of performing data generation in multilingual, low-resource settings for a challenging task of commonsense reasoning. Although the general-purpose LLMs exhibit impressive zero-shot learning capabilities, smaller task-specific models can often be more accurate, cost-effective, and practical, given that enough training data is available. Therefore, improving the accuracy of task-specific models by generating synthetic data for fine-tuning is still an important research topic.

**Reasons To Reject:**

One of the weaknesses of the presented approach is that it does not provide a mechanism to control the diversity of the synthesized examples. Although the authors claim that their method generates diverse examples (lines 61-62), it has neither been supported empirically nor discussed in the paper. Notably, the results presented in Section 5.3 suggest that the benefits of generating larger amounts of data are rather limited, possibly due to the low diversity of the synthesized examples.

In addition, the clarity of the paper needs to be improved. Specifically, the authors generated 2-4k new data points to fine-tune the models for each data set. It is unclear whether all instances were synthesized in one inference step or the models were prompted multiple times for each data set. Moreover, given that the number of valid examples obtained with different models differs substantially (see the success rates in Table 3), it needs to be clarified why an equal number of synthesized data points is reported for each LLM used for generation. Presumably, the models were prompted multiple times until the required number of examples was reached, but the paper needs to explain this. Generating small batches of examples by prompting the model multiple times, each time using a different set of seed examples, might also improve the diversity and, in turn, the quality of the synthesized data.

Another point is that the choice of the number of synthesized samples per data set should be discussed in the paper. The number of examples is neither fixed nor proportional to the size of the corresponding original data set. For XWinograd, which has the largest training set out of the examined benchmarks, additional 2k examples are generated, which doubles its size. In contrast, the size of XCOPA and XStoryCloze increases 10 and  6.7 times, respectively.

Finally, the authors need to discuss their method's effectiveness in comparison with other data augmentation approaches. Note that after deduplication and discarding invalid instances, only up to 42% of examples are retained (in the case of open-access models). The usage of closed generative LLMs might also not be cost-effective. The reader would benefit from a more in-depth discussion of the advantages and disadvantages of the proposed method. The limitations section needs to be updated accordingly.

**Reproducibility:**

4: Could mostly reproduce the results, but there may be some variation because of sample variance or minor variations in their interpretation of the protocol or method.

**Reviewer Confidence:**

4: Quite sure. I tried to check the important points carefully. It's unlikely, though conceivable, that I missed something that should affect my ratings.

---

> ### Author Rebuttal · Authors · 2023-08-28
>
> We appreciate the comprehensive feedback and suggestions from the reviewer. We would like to first address the main concern related to the diversity and the data augmentation approach. Our *clarification* on the data generation process is as follows:
>
> Our **data generation process** involves the following steps:
> 1. We set a total number of examples we want to generate (usually related to budget), in our case it is around 3K (but it can be a fixed number of data or a fixed ratio w.r.t the original dataset)
> 2. Generate examples as follows until we have enough valid examples:
>
>     a. To ensure *diversity*, we **randomly** sample **n** examples (*typically set to 5*) each time from the train sets;
>
>     b. Append the sampled examples to the instruction, and prompt the model to generate **m** (*set to 5-10*) new examples;
>
>     c. Post processing and add the *valid and unique* examples to the generated set.
>
> The values of *m* and *n* depend on example length and model constraints (input/output length). In our case, we set them to 5-10 depending on the specifics of each dataset.
>
> We want to clarify that we employ random samples in the prompt each time, in line with the reviewer's suggestion.
> We acknowledge the need to explicitly state that examples are not generated all at once and to *specify m and n values*. This clarification will be incorporated in the camera-ready version.
>
> Beyond addressing the data augmentation approach, we address additional concerns:
> 1. Regarding diversity, a study in the appendix (lines 860-862) compares topic coverage to evaluate diversity through "randomly sampling n examples from the dataset in the instructions." While this isn't emphasised presently, we intend to enhance this description for the camera-ready version.the current version and will improve the description in the camera-ready version.
> 2. Number of generated examples. We initially focus on fixed-budget experiments (e.g., generating 3K examples). We acknowledge the valuable suggestion of comparing experiments with a fixed data ratio. To this end, we've conducted experiments on GPT-4 generated English XCOPA data, inclusive of the original 400 train data. Our plan is to extend these experiments to cover additional datasets and expand findings in the camera-ready version.
>
> | XCOPA-EN |   1X   |  2X   |  5X   | 10X   |
> |-----------|------|-------|------|-------|
> | Data Size  |   400    | 800| 2000| 4000|
> | mBERT     |   62.6   |  68.4  |  69.2 |  70.4 |
> | XLMR-base  |   57.6   |  59.4  |  66.4 |  69.8 |
>
> 3. Regarding other data augmentation approaches, methods that focus on the selection and filtering of the automatic generation data  can also be applied after our augmentation pipeline; our work aims to explore the potential of data augmentation from LLMs, driving exploration in LLMs and open-source models like LLAMA2. We will expand the comparison and the limitation section accordingly.
>
> We would also like to answer the **questions**:
> 1. We set m and n to 5-10 in our experiments
> 2. We use the default temperature (1.0)
>
> We kindly request the reviewer's consideration to revise the score in light of our provided clarification.

---

### Official Review · Reviewer_4SYH · 2023-08-08

**Soundness:** 4

**Excitement:**

4: Strong: This paper deepens the understanding of some phenomenon or lowers the barriers to an existing research direction.

**Paper Topic And Main Contributions:**

The paper explores the use of LLMS to generate additional training data for common sense reasoning in cross-lingual setting. The authors compare different LLMs, tasks, and languages. They experiment both with generating data in English and in the target language. They also present a human evaluation of the quality of the target-language generated data.

The main contributions of the paper are:

- empirical analysis of the impact that synthetic data has on the performance of downstream classifiers
- a systematic comparison between different models, datasets, and languages
- the resources created as part of the research

**Questions For The Authors:**

In table 4, the number of generated examples is independent of the number of original examples. Did you consider running experiments with a fixed ratio of synthetic examples (e.g. exactly the same size as original examples, 2x, 3x) or with a fixed "total" number of examples (e.g. 1000, 2000...)?

When analyzing results on TA, you find that only few examples can help improve the performance of the model to a degree. Did you run similar examples with other languages using a few (e.g., 5, 10, 50) examples?

**Reasons To Accept:**

- well written paper, with well defined methodology and strong evaluation
- promising experimental results
- resources and findings that are useful to the community

**Reasons To Reject:**

None, it's a very solid paper.

**Reproducibility:**

5: Could easily reproduce the results.

**Reviewer Confidence:**

4: Quite sure. I tried to check the important points carefully. It's unlikely, though conceivable, that I missed something that should affect my ratings.

**Typos Grammar Style And Presentation Improvements:**

I suggest that you make stronger claims in the abstract/introduction and bring some quantitative results (e.g. improvement by X % in best case)

---

> ### Author Rebuttal · Authors · 2023-08-28
>
> We thank the reviewer for the encouraging feedback, questions and suggestions. We would like to address the questions as follows:
>
> We initially focus on fixed-budget experiments (e.g., generating 3K examples). We acknowledge the valuable suggestion of comparing experiments with a fixed data ratio. To this end, we've conducted experiments on GPT-4 generated English XCOPA data, inclusive of the original 400 train data. Our plan is to extend these experiments to cover additional datasets and expand findings in the camera-ready version.
>
> | XCOPA-EN |   1X   |  2X   |  5X   | 10X   |
> |-----------|------|-------|------|-------|
> | Data Size  |   400    | 800| 2000| 4000|
> | mBERT     |   62.6   |  68.4  |  69.2 |  70.4 |
> | XLMR-base  |   57.6   |  59.4  |  66.4 |  69.8 |
>
> The observation that only a few examples in TA contribute to the performance is derived from the human evaluation quality of TA. We have not run similar experiments on other languages, but it is a great suggestion to study and add to the final version.
>
> We will also add the quantitative improvement results in the abstract as suggested to strengthen the claim.

---

### Meta-Review · Area_Chair_4a26 · 2023-09-16

**Recommendation:** 5

**Metareview:**

This paper explores the used of several LLMs (like StableVicuna and GPT-4) like  to generate synthetic data to train a smaller multilingual language models like mBERT and XLM-R. They performed evaluation on several languages and on three tasks: XCOPA, XWinograd and XStoryCloze.

All the reviewers found this to be interesting and a strong evaluation especially for low-resource settings and the efficiency of smaller models. It's also great the authors compared with machine translated texts which had been done in past works. There a few comments by the reviewers about the diversity of the generated sentences, the use of bigger models, use of more recent LLMs for generation, and some unclear part of the write-up, which the authors have adequately addressed. I believe this paper is a good contribution of LLMs for non-English languages especially in terms of building smaller models with similar or better accuracy than LLMs.

---

### Decision · Program_Chairs · 2023-10-07

**Decision:**

Accept-Main

**Comment:**

This paper explores the used of several LLMs (like StableVicuna and GPT-4) like  to generate synthetic data to train a smaller multilingual language models like mBERT and XLM-R. They performed evaluation on several languages and on three tasks: XCOPA, XWinograd and XStoryCloze.

All the reviewers found this to be interesting and a strong evaluation especially for low-resource settings and the efficiency of smaller models. It's also great the authors compared with machine translated texts which had been done in past works. There a few comments by the reviewers about the diversity of the generated sentences, the use of bigger models, use of more recent LLMs for generation, and some unclear part of the write-up, which the authors have adequately addressed. I believe this paper is a good contribution of LLMs for non-English languages especially in terms of building smaller models with similar or better accuracy than LLMs.